# NoLoCo: No-all-reduce Low Communication Training Method for Large Language Models

## Abstract

Training large language models is generally done via optimization methods on clusters containing tens of thousands of accelerators, communicating over a high-bandwidth interconnect. Scaling up these clusters is expensive and can become impractical, imposing limits on the size of models that can be trained. Several recent studies have proposed training methods that are less communication intensive, avoiding the need for a highly connected compute cluster. These state-of-the-art low communication training methods still employ a synchronization step for model parameters, which, when performed over all model replicas, can become costly on a low-bandwidth network.

In this work, we propose a novel optimization method, NoLoCo, that does not explicitly synchronize all model parameters during training and, as a result, does not require any collective communication. NoLoCo implicitly synchronizes model weights via a novel variant of the Nesterov momentum optimizer by partially averaging model weights with a randomly selected other one. We provide both a theoretical convergence analysis for our proposed optimizer as well as empirical results from language model training. We benchmark NoLoCo on a wide range of accelerator counts and model sizes, between 125M to 6.8B parameters. Our method requires significantly less communication overhead than fully sharded data parallel training or even widely used low communication training method, DiLoCo. The synchronization step itself is estimated to be one magnitude faster than the all-reduce used in DiLoCo for few hundred accelerators training over the internet. We also do not have any global blocking communication that reduces accelerator idling time. Compared to DiLoCo, we also observe up to $4\%$ faster convergence rate with wide range of model sizes and accelerator counts.

## 1 Introduction

Large language models (LLMs) have recently shown impressive performance on a wide variety of tasks, including natural language understanding (Liu et al., 2024; Team et al., 2023; Touvron et al., 2023; Zhang et al., 2022); image related tasks (Zhang et al., 2025; Gao et al., 2024; Zhu et al., 2023; Lin et al., 2023); or speech recognition and generation (Maiti et al., 2024; Gourav et al., 2024; Rubenstein et al., 2023; Xu et al., 2025). These large models are usually trained by a combination of different distributed training methods such as data parallelism (Rasley et al., 2020), pipeline parallelism (Huang et al., 2019; Narayanan et al., 2021; Sun et al., 2024), and others (Shoeybi et al., 2019; Rasley et al., 2020; Liu et al., 2023; Shyam et al., 2024; Fujii et al., 2024; Liu et al., 2024; Cai et al., 2024). The aforementioned training methods are bandwidth intensive and require a high-bandwidth interconnection fabric available between individual compute nodes that is generally only available in dedicated machine learning clusters (Team et al., 2023; Grattafiori et al., 2024; Duan et al., 2024; Intelligence, 2024). This requirement increases the cost of training and poses a limit on the training scale as highly connected computer clusters cannot be scaled easily beyond a data center.

Recently, a number of studies have aimed to address this issue by proposing algorithms that scale better than traditional distributed training algorithms in low-bandwidth and high latency network (Douillard et al., 2023; 2024; Ryabinin et al., 2023; Li et al., 2022; Biswas et al., 2024; Kale et al., 2025; Charles et al., 2025). Most of these methods use an explicit step to synchronize the data parallel instances of the model during training, typically using all-reduce operations. This synchronization step can

take several minutes in highly distributed network and lead to poor overall scaling efficiency of the training algorithm (Jaghouar et al., 2024; Yuan et al., 2022).

The main contribution of the paper is a novel optimization method, NoLoCo, for training LLMs that does not use any explicit all-to-all communication. NoLoCo is built upon the inner-outer optimizer paradigm together with the idea of epidemic learning where averaging is done among subset of accelerators instead of all. Specifically, in NoLoCo, outer synchronization is done via only pairs of the accelerators, rather than all of them. Moreover, each inner optimizer step is done via random pipeline routing of accelerators which implicitly helps different pipelines to converge with less synchronisation. Furthermore, we modify the Nesterov momentum optimizer to prevent the weights of the same stage from diverging. We also provide a convergence proof for our modified optimizer.

We test our method with a state of the art low communication method, DiLoCo, and with a traditional distributed model training in the language modeling task with two datasets (Pushshift Reddit and C4) and several model sizes (125M, 1.3B and 6.8B parameters). Our experimental results show that our method is more communication efficient and also converges up to $4\%$ faster than DiLoCo for wide range worker counts, and model sizes. The speed-up from omitting the all-to-all communication increases with increasing number of workers and network latency variance. Source code for running the experiments is available in GitHub.

## 2 RELATED WORK

### 2.1 DECENTRALIZED TRAINING METHODS

The common data-parallel optimization algorithm keeps all model parameters synchronized across data-parallel workers by always performing an all-reduction on the gradients before the optimizer step (Rasley et al., 2020). Decentralized training methods such as DiLoCo have relaxed this assumption by allowing the model parameters to diverge during local steps and only synchronizing them at the outer optimizer steps, performed less frequently (Douillard et al., 2023). Specifically, DiLoCo and its variations (Douillard et al., 2024; Li et al., 2022; Biswas et al., 2024; Kale et al., 2025; Peng et al., 2024; Charles et al., 2025; Qi et al., 2025) divide the optimization process into inner and outer optimizer steps similar to Lookahead optimizer (Zhang et al., 2019). During the inner optimizer steps, only local weights are updated, and different copies of the model can have different weights. The outer optimization step uses local weights to compute an outer gradient that is applied to update global weights shared between all copies of the model (Douillard et al., 2023).

DiLoCo greatly reduces the frequency of all-reduces compared with regular data parallel training as they are only performed during the outer optimizer steps as opposed to every optimizer step. When accelerator connections are slow or there are enormous number of accelerators, the DiLoCo outer optimizer steps can consume a significant amount of time (Jaghouar et al., 2024; Yuan et al., 2022). Recent studies have aimed to address this by staggering the all-reduce communication with compute (Kale et al., 2025; Qi et al., 2025; Douillard et al., 2025) or only synchronizing a subset of model parameters (Biswas et al., 2024). Both of these ideas can be extended to NoLoCo to further reduce the communication overhead.

Methods using outer optimizer steps (Douillard et al., 2023; Kale et al., 2025; Biswas et al., 2024) still require all-to-all communication during the outer optimizer steps. To overcome this, several studies in the federated learning paradigm (Hegedűs et al., 2019; De Vos et al., 2023; Du et al., 2024; Dhasade et al., 2025; Sad et al., 2025) have proposed various training methods that aim to avoid explicit all-to-all communication by replacing it with local communication. For example, epidemic learning (De Vos et al., 2023) proposed the use of local averaging to synchronize the subset of weights during training instead of all weights to avoid all-reduces.

### 2.2 DYNAMIC PIPELINE ROUTING

Pipeline parallelism (Huang et al., 2019; Narayanan et al., 2021; Sun et al., 2024) is a popular choice for model parallelism in a low bandwidth environment as it requires less network bandwidth and involves less blocking communication than the fully sharded data parallel training with ZeRO-3 (Rasley et al., 2020) or the tensor parallelism (Shoeybi et al., 2019). In pipeline parallel training, model is split to consecutive stages where each stage passes it's compute outputs to the next stage and

receive inputs from the prior stage (Huang et al., 2019). During forward pass model stages have to wait for inputs from the prior stage and during backward pass for gradients from the subsequent stage. This leads to formation of the computation bubble where certain devices will be inherently idle.

Swarm (Ryabinin et al., 2023) is one of the earliest works allowing pipeline stages receive inputs from arbitrary replicas of the prior model stage and vice-versa. In this setting, a model stage can start computing immediately once inputs are available from any replica of the prior stage as opposed to waiting for a dedicated model replica, which is the case for regular pipeline parallelism. The approach effectively reduces blocking communication and renders itself well for load balancing (Ryabinin et al., 2023). In SWARM, pipeline routing is done based on the load balancing using a message queue like setup. For equal workers and uniform network topology, this becomes essentially random routing. In this study, we will employ random routing as it is a good proxy for the SWARM routing process from optimizer convergence point of view.

Later DiPaCo (Douillard et al., 2024) proposed similar setup to SWARM, but used an explicit routing model loosely related to mixture-of-expert (MoE) parallelism (Liu et al., 2024; Cai et al., 2024). The main difference to between DiPaCo and the MoE parallelism is that in DiPaCo the routing is done at the sample level while MoE parallelism is generally done at the token level; and that the routing model in DiPaCo is a separate model while in MoE models it is part of the model itself (Cai et al., 2024). The routing model in DiPaCo also need to be trained before the actual training.

DiPaCo can theoretically produce less correlated outer gradients due to the router, which is desirable for estimating the expected value using sample means. However, it can also lead to load balancing issues similar to standard MoE parallelism (Cai et al., 2024). In addition, having different token counts used within the inner optimizer steps by different workers can degrade outer gradient estimates. DiPaCo aimed to address this by using weighted averaging where the weights are given as the ratio of tokens used by the worker and all tokens used by all workers.

## 3 NoLoCo

NoLoCo utilizes Data Parallelism (DP) and Pipeline Parallelism (PP) methods with the following modifications: (i) for inner optimization step of PP, at each iteration, different pipeline paths are chosen among the accelerators rather than having a fixed path, (ii) for outer optimization step of DP, only pairs of accelerators are synchronized rather than all. To prevent the weights of the same stage from diverging, we modify the Nesterov momentum optimizer step.

### 3.1 Inner optimizer step via dynamic pipeline routing

NoLoCo uses dynamic PP where an accelerator can receive inputs from any instance of the prior pipeline stage and forwards outputs to any instance of next pipeline stage. We use random permutations to group workers, and perform the routing based on the random groups that guarantees good load balancing. This is illustrated in Fig. 1A. During the backward pass, gradients follow the same path that was chosen during the forward pass.

The random routing allows mixture of the weights of different DP components as their inputs and outputs might be used by the same model pipeline stages. We hypothesize this creates an implicit effect to drive the weights of different DP models closer without requiring frequent synchronization.

### 3.2 Outer optimizer step with modified Nesterov momentum

We aim to relax the outer synchronisation even further by synchronizing the model parameters with a smaller local group, rather than the all-to-all reduction performed in DiLoCo-like methods.

Let us assume there are $N$ model instances in the whole network, and we synchronize among a randomly chosen local subgroup of $n$ instances where $N \gg n$. For each iteration we update the local subgroup to obtain information from different workers. In the local subgroup, we have $n$ model instances at step $t$, $\phi_{t,i}$, where $i < n$ indicates the model instance. We will refer to $\phi_{t,i}$ as slow weights as in the Lookahead optimizer (Zhang et al., 2019). To progress to the next step, we perform $m$ local optimizer steps on $\phi_{t,i}$ to obtain fast model weights $\theta_{t+1,i}$. This step is the same to the inner optimizer steps in DiLoCo and similar to Lookahead optimizer inner steps. We use the fast model

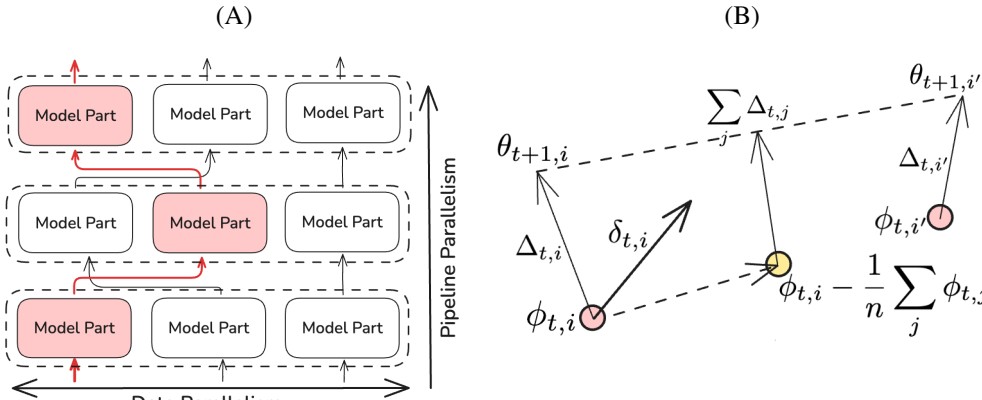

Figure 1: (A): Illustration of dynamic PP routing with DP. Model is split to consecutive pipeline stages (shown vertically), and each stage is replicated to process data in parallel (shown horizontally). (B): Illustration of different terms of the outer momentum term for group consisting of two workers. Red dots show the current slow weights. Yellow dot shows the average of the slow weights.

weights to compute a local outer gradient:

$$\Delta_{t,i} = \theta_{t+1,i} - \phi_{t,i}. \tag{1}$$

We modify the expression for Nesterov momentum by computing it over a local group as opposed to all data parallel workers and introduce a term to account for difference in the local slow weights:

$$\delta_{t,i} = \alpha \delta_{t-1,i} - \frac{\beta}{n} \left( \sum_j \Delta_{t,j} \right) - \gamma \left( \phi_{t,i} - \frac{1}{n} \sum_j \phi_{t,j} \right), \tag{2}$$

where $\alpha$ is Nesterov momentum parameter; $\beta$ is the outer learning rate; $\gamma$ is a local weight averaging parameter. The expected values are taken over a random sub groups of size $n$. If the sub group consists of all model instances, Eq. 2 simplifies to regular DiLoCo outer optimizer momentum and the last term diminishes: $\delta_t = \alpha \delta_{t-1} - \frac{\beta}{n} \left( \sum_j \Delta_{t,j} \right)$. The third term can also be viewed as a rolling average over weight differences between the worker and random workers.

Finally, local weights are updated by the momentum $\delta_{t,i}$ in the same way as in Lookahead optimizer:

$$\phi_{t+1,i} = \phi_{t,i} + \delta_{t,i}. \tag{3}$$

Fig. 1B illustrates relationship between different terms in the method for outer gradient computation involving two workers.

In our experiments, we use the minimum group size for random subgroups, which is $n = 2$. During the outer optimizer step, workers have to share their outer gradients (Eq. 1) and prior slow weights $\phi_{t,i}$. The prior slow weights can be communicated already at the end of the prior outer step. This allows overlapping communication of the slow weights with the computation for the next fast weights.

Intuitively, the method should have convergence properties very similar to those of DiLoCo as the mean term in Eq. 2 is a rolling average of the slow weights over the duration of training. We prove this by showing that the modified Nesterov optimizer given by Eq. 2 converges to the optima $\theta = 0$ for a quadratic loss of the form $\mathcal{L}(\theta) = \frac{1}{2}(\theta - c)^{\mathrm{T}} \mathrm{A}(\theta - c)$, where $c \sim \mathcal{N}(0, \Sigma)$ with a constant covariance matrix $\Sigma$, and A is a positive definite symmetric matrix (Schaul et al., 2013; Wu et al., 2018; Zhang et al., 2019). We also assume that the inner optimizer uses stochastic gradient descent with a constant learning rate $\omega$. With these assumptions following theorem hold:

**Theorem 1** *When the outer iteration step count $t \to \infty$, the expected value of the slow weights $\mathbb{E}(\phi_{t,i}) \to 0$, and the variance $\mathbb{V}(\phi_{t,i}) \propto \omega^2$.*

Proof of the above theorem is given in Appendix A. The variance part of the theorem also hints that the inner learning rate - learning rate schedules in particular - can be an effective way to control

| Parameter Name | Small | Medium | Large |
|---|---|---|---|
| Hidden size | 768 | 2048 | 4096 |
| Layers | 12 | 24 | 32 |
| Intermediate size | 3072 | 8192 | 16,384 |
| Attention heads | 16 | 32 | 32 |
| (Inner) Learning-rate | 0.0006 | 0.0002 | 0.00012 |
| Global batch size | 0.5M | 1M | 2M |
| Transformer Parameters | 125M | 1.3B | 6.8B |

Table 1: Summary of model hyper-parameters. Batch sizes are expressed in tokens.

the divergence of the weights during the training. One can initiate the training with a large learning rate and decay it towards end of the training to essentially obtain a very tight cluster of models. We present empirical evidence for this in Section 5.

## 4 EXPERIMENTAL SETUP

We study the optimization methods in the context of next token prediction task. We use two datasets, pushshift reddit data (Baumgartner et al., 2020) and C4 (Dodge et al., 2021) to probe the training approaches. For benchmarking, we use 10 million tokens hold out from training data for reddit and the validation partition for C4. All data is tokenized by the Llama sentencepiece tokenizer with a vocabulary size of $128,000$ tokens and formatted to sequences of 1024 tokens.

Global batch size and learning rate are taken from studies Shuai et al. (2024); Zhang et al. (2022). Each run has a linear warm-up of 1000 steps and cosine learning rate schedule applied after the warm-up period to reduce the learning rate by one magnitude compared to the maximum learning rate. All training runs are done over $25,000$ optimizer steps. We explore 3 model sizes: small, medium and large Llama models with 125M, 1.3B and 6.8B parameters respectively, and all models have the same vocabulary size of $128,000$ tokens. Model hyper-parameters are outlined in Table 1.

We use Adam as the (inner) optimizer for all experiments and applied gradient clipping for gradients larger than unity. Both methods use the same outer learning rate, $\beta = 0.7$. For DiLoCo we use a momentum value of $\alpha = 0.3$ that was found to produce better results in our setting than standard $\alpha = 0.9$; and we apply the outer optimizer step at every 100 steps. For NoLoCo, we use a higher momentum value of $\alpha = 0.5$; the group size of two workers; and employ the optimizer step every at 50 steps. It's note worthy that with the doubled frequency of outer optimizer steps, NoLoCo still requires much less communication than DiLoCo since $N \gg 2 \cdot n := 4$. All computations are done in bfloat16 numerical precision and multi-head attentions are computed using the flash-attention (Dao et al., 2022). Finally, the source for running the experiments is available in GitHub.

## 5 RESULTS AND DISCUSSION

### 5.1 TRAINING RESULTS

Validation perplexities at the end of the training are shown in Table 2 for both Reddit and C4 datasets. We observe that both methods are slightly worse than fully sharded data parallel (FSDP) training for all the cases considered, typically few percent worse, but in some cases even $30\%$ (C4, Small model, 16 data parallel world size). The perplexity gap is generally larger with smaller models and large data parallel world sizes; and decreases when the model size becomes larger or the data parallel world size becomes smaller. This finding is consistent with the findings of recent study in DiLoCo scaling (Charles et al., 2025). In this regard, both NoLoCo and DiLoCo convergence scale similarly respect to data parallel world size and model size.

The perplexity numbers presented here are subject to the training hyperparameters. In particular, the batch size and learning rate were chosen from a study optimized for FSDP and hence are likely sub-optimal for both DiLoCo and NoLoCo. We found that increasing the batch size improved the training method performance (see Appendix C). However, extensive hyper parameter search for all

| Model | Total | DP | PP | Pushshift Reddit | | | C4 | | |
|---|---|---|---|---|---|---|---|---|---|
| | | | | FSDP | DiLoCo | NoLoCo | FSDP | DiLoCo | NoLoCo |
| Small | 8 | 8 | 1 | 25.5 | 27.6 | **27.3** | 29.1 | 35.4 | **34.5** |
| | 8 | 4 | 2 | 25.7 | 26.8 | **26.4** | 29.1 | 32.1 | **31.3** |
| | 16 | 8 | 2 | 25.5 | 27.6 | **27.2** | 29.1 | 34.0 | **33.1** |
| | 32 | 16 | 2 | 25.5 | 29.7 | **29.1** | 29.1 | 39.1 | **37.7** |
| Medium | 16 | 8 | 2 | 19.6 | 21.0 | **20.5** | 18.8 | 21.8 | **21.1** |
| | 32 | 16 | 2 | 19.6 | 21.2 | **20.7** | 18.8 | **23.2** | 23.4 |
| | 64 | 16 | 4 | 19.6 | 21.0 | **20.9** | 18.8 | **22.6** | 22.9 |
| Large | 64 | 16 | 4 | 16.1 | 18.0 | **17.5** | 15.7 | 17.3 | **16.6** |

Table 2: Validation perplexity values for different world sizes and models at the end of the training. DP stands for the data parallel world size and PP for the number of pipeline stages. FSDP stands for fully sharded data parallel training. Best perplexity results are highlighted with a bold font.

the methods and model sizes is beyond scope of this study and we report most of the results using hyper-parameters from the study of Zhang et al. (2022).

Comparing our method with DiLoCo, we observe that NiLoCo is slightly better than DiLoCo in most experiments (all reddit experiments, and most experiments in C4). This seems counterintuitive, as one would expect the variations in model weights during the training to slow the convergence, not improve it. We hypothesize that this could be due to small perturbations in model weights having a regularization effect on training. All training data batches in this study were within the first epoch, but larger datasets have been shown to contain similar text sequences that can cause slight over-training already within the first epoch. This could also explain why we observe better performance in the pushshift reddit data as opposed to C4 which has more variety of topics and hence less likely to experience some level of over-fitting within the first epoch.

We also compared the methods in a pure data parallel setting without the PP and the random routing. We observe that for NoLoCo there was a minor degradation in the final perplexity and the convergence rate was slightly slower compared with the PP case. For DiLoCo, the opposite was true: perplexity was unchanged but we did note a negligible increase in the convergence rate (not shown). We conclude that the random routing has a minor impact on the convergence rate.

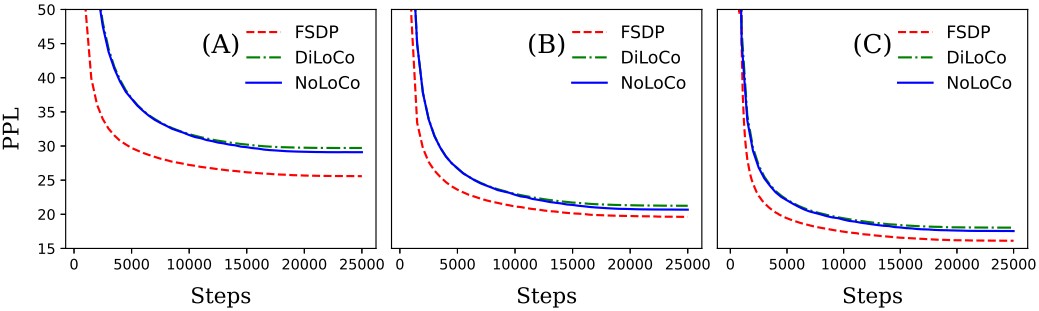

Figure 2: Reddit validation perplexity of different training methods at different optimizer step counts. Solid blue curve is NoLoCo, dashed red curve show FSDP, and dashed green curve is DiLoCo. (A): Small model with DP world size of 8 and two pipeline stages; (B): Medium model with DP world size of 8 and two pipeline stages. (C): Large model with DP world size of 16 and four pipeline stages.

Fig. 2 show validation perplexity over the course of training for all model sizes (see Appendix D for corresponding training loss). We can observe that the gap between the FSDP and the decentralized methods becomes less with model size and that NoLoCo has slightly lower perplexity towards the end of the training. This can be easily observed in Fig. 3A that shows the convergence of the relative validation perplexity difference between DiLoCo and NoLoCo for Reddit data. The shown relative perplexity difference is computed by

$$\text{Rel. PPL Diff} = \frac{\text{DiLoCo Perplexity} - \text{NoLoCo Perplexity}}{\text{FSDP Perplexity}}. \tag{4}$$

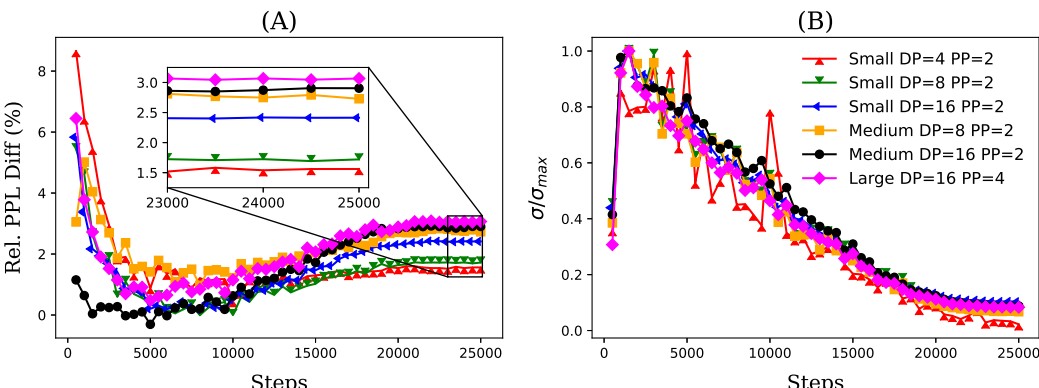

Figure 3: (A): Relative validation perplexity difference between DiLoCo and NoLoCo. Perplexity numbers are normalized by the FSDP perplexity at the same optimizer step count. Positive number indicate faster convergence compared to DiLoCo. (B): Standard deviation of the model weights across the data parallel world size normalized by the largest value within the training run.

We find that NoLoCo has typically a few percent lower perplexity during the beginning and the end of the training while they are fairly close in the intermediate stage of training. The difference in the late stage perplexity depended on the dataset. For pushshift reddit, we observe that NoLoCo is converging faster at the late stage while for C4 the results are varying and depended on the model size and the data parallel world size.

Finally, Fig. 3B shows convergence of the different model replicas during the training. We observe that the standard deviation between different data parallel model replicas peaks after the warm-up and converges throughout the training. Theorem 1 suggests that the model instance variance is proportional to the square of the inner learning rate once convergence is reached. We find empirically that the Pearson correlation coefficient of the standard deviation and the learning rate ranges indeed between 0.91 and 0.97 suggesting that the weight variance across different instances is controlled to a high degree by the inner learning rate as predicted by the theorem. Hence, learning rate scheduler can be used effectively to obtain eventual consistency of the weights in NoLoCo.

## 5.2 Latency Analysis

We will derive theoretical speed-up for the NoLoCo local averaging compared with a tree all-reduce. We consider $n$ workers with each having a message send time of $t_c$ to any other worker. Tree all-reduce is composed of two stages: a reduce to the root of the tree followed by a broadcast from the root of the tree to all leave nodes. The total time that it takes to do this will: $t_{all} \approx 2t_c \log_2(n)$.

For the local averaging with groups of two, the overall time is simply $2t_c$ and hence the ratio will be $\sim \log_2(n)$. This equation ignores the fact that not all communication takes the same time and in practice the communication time follows a distribution. We model this by assuming that the communication time $t$ follows a log-normal distribution $t \sim \text{LogNormal}(\mu, \sigma^2)$. The time it takes for a parent node to receive message from it's children is the maximum of the children nodes' sending time: $t_{local} = \max(t_1, t_2)$. If $t_1$ and $t_2$ are independent identically distributed log-normal random variables, the expected value of $t_{local}$ is given by:

$$\mathbb{E}(t_{local}) = \left(1 + \text{erf}(\frac{\sigma}{2})\right) \exp\left(\mu + \frac{\sigma^2}{2}\right). \quad (5)$$

Also, $2\mathbb{E}(t_{local})$ is the mean time it takes for the local averaging. Fig. 4A show the ratio of tree-reduce expected time to local averaging expected time in terms of different world sizes and message sending time variances. We can see that the tree-reduce slows significantly when the message sending time standard deviation increases, which is a typical case for public networks. This effect becomes larger for the larger world sizes as expected.

The above analysis also assumes that all the workers call the all-reduce operation at the same time, but this is in practice not true and workers would arrive to the communication call at different

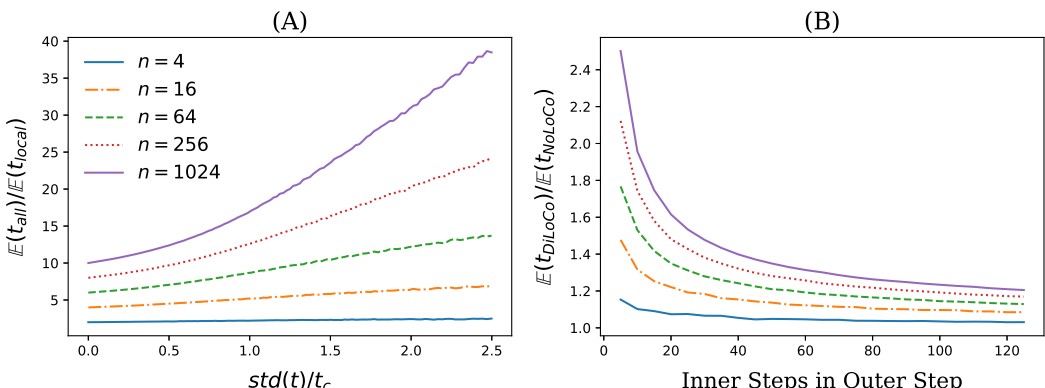

Figure 4: (A): Simulated ratio tree-reduce expected time to local averaging expected time. $n$ denotes here a world size and $t_c = \exp(\mu + \sigma^2/2)$ is the expected time for sending single message. (B): Simulated ratio of total training times between DiLoCo and NoLoCo without explicit communication overhead. Each training run consisted of 500 outer optimizer steps and variable number of inner optimizer steps. The inner optimizer step latency is modeled as log-normal distribution with $\mu = 1$ and $\sigma^2 = 0.5$. All-reduce and local-reduce communication times are assumed to be negligible.

| Method | 100 Mb/s | 1 Gb/s | 100 Gb/s |
|--------|---------:|-------:|---------:|
| FSDP   | 5368.41 s | 548.97 s | 41.39 s |
| DiLoCo | 61.80 s | 6.55 s | 1.70 s |
| NoLoCo | 15.26 s | 2.80 s | 1.64 s |

Table 3: Average time for (inner) training steps for all methods with varying GPU interlink bandwidths. All results are obtained from using 32 GPUs; medium model size; and no PP. Bandwidths limits are applied for both download and upload speeds, and are expressed in bits per second. Averaging is taken over two outer steps; and outer steps consisted of hundred inner steps.

times. We performed similar analysis by modeling the time each inner optimizer step takes as log-normal distribution and observed how long it will take for all processes to finish 250 outer iterations. We omitted the local averaging and all-reduce time to highlight the effect of the global blocking communication present in DiLoCo (and it's variations), but not in NoLoCo. Results are shown in Fig. 4B. One can observe that the difference between NoLoCo and DiLoCo increases with increasing world size and is $\sim 20\%$ for 100 inner steps within an outer step using 1024 accelerators. Performing outer optimizer steps more often increases the overhead. This overhead originating from the blocking communication is present in addition to the overhead originating from all-reduce latency.

To give reader a better understanding how these effects manifest in actual training, we limited network bandwidths between different workers and executed two (outer) training steps in a network consisting of 32 workers for all methods. We used the medium size model with pure DP and three different interlink speeds in out experiments. The slowest interlink speed (100Mb/s) was characteristic for regular consumer grade internet; the middle speed (1Gb/s) can be found from high grade consumer internet connections; and 100Gb/s represents interlink speeds found in data centers.

Table 3 shows the results from these experiments for different methods. We can see significant difference between regular DP training speed and decentralized methods even with $100\,\mathrm{Gb/s}$ interlink speed. The ratio of NoLoCo and DiLoCo training step times approaches the theoretical limit of $\log_2(32) = 5$ as the network speed is reduced. This is expected with the lower network speeds since the all-reduce becomes dominant part of the overall training time.

### 5.3 EFFECTS OF RANDOM PIPELINE ROUTING

To analyze the effect of random routing in the PP communication, we perform an ablation study comparing random routing with fixed routing. For the fixed routing, nodes only send and receive values from a fixed prior and subsequent model stage, as in typical setups Huang et al. (2019);

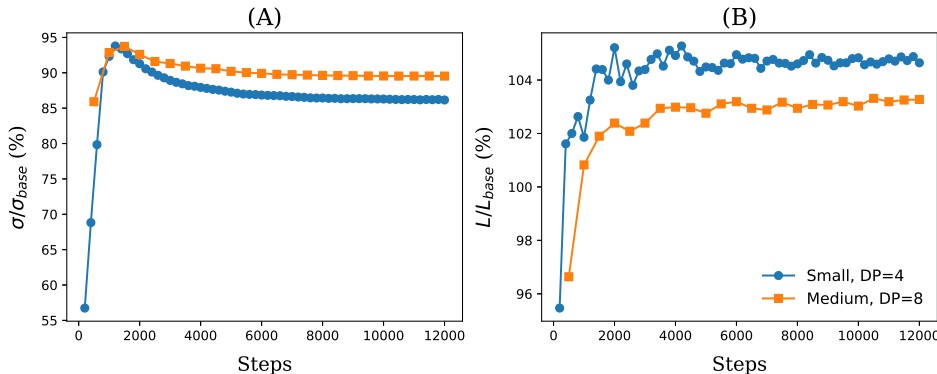

Figure 5: Results from training without any outer optimizer steps. The baseline loss and model replica variance is computed from DP number of independent runs. The combined training is only using the random pipeline routing. Figures shows the ratio of weight standard deviations (A) and validation perplexities (B) between the random pipeline routing and without it.

Narayanan et al. (2021); Sun et al. (2024). We also remove the outer optimizer synchronization (for both routing methods), thus nodes in the same stage never exchange information directly. Thus, without random routing the setup is the same as running separate training jobs without DP. Using this setup we repeat experiments for Reddit using the small and medium model sizes.

We present the results in Fig. 5. We observe that for small model the standard deviation is $\sim 15\%$ lower than in the same run without random routing between different PP pipelines. This effect becomes less pronounced for the medium model with larger DP world size, namely $\sim 10\%$ lower standard deviation. Thus we observe that through the PP training, nodes in the same stage implicitly exchange information about their weights, without directly synchronizing. We attribute the faster convergence observed in Fig. 2 to this fact.

## 6 SUMMARY

We proposed a novel low communication decentralized training method, NoLoCo, that requires only subgroup synchronization in outer optimizer steps and avoids collective all-to-all communication. While reducing the synchronisation group, to prevent diverge of the model weights, we introduced a modification on the Nesterov optimizer used in the outer step. We provide a convergence proof of NoLoCo as well as show that its convergence rate is comparable with fully synchronized DP methods.

We also compared NoLoCo with a less frequently synchronizing method (DiLoCo) via various model sizes ranging from 125M to 6.8B parameters; two different language modeling datasets (C4 and pushshift reddit); and a number of parallel worker counts. We found that NoLoCo converges up to $4\%$ faster than DiLoCo in our experiments while not using any all-to-all communication. We hypothesize that this is because of the regularization effect coming from slight variations in different instances.

Speed-up from removing the all-to-all communication outer optimizer steps depends on the standard deviation of the message sending latency and logarithmically on the DP worker count. We confirmed this behavior in our experiments where we observed two and four times faster training step times with NoLoCo compared with DiLoCo using $1\,\mathrm{Gb/s}$ and $100\,\mathrm{Mb/s}$ interlink speeds, respectively.

NoLoCo - unlike DiLoCo - produces an ensemble of models as the weights are not explicitly synchronized. We found that the standard deviation of the model weights across different instances is controlled to a high degree by the inner learning rate. Hence, a learning rate scheduler can be used effectively to obtain eventual consistency of different model instances.

We have demonstrated that local averaging is a viable option for training models with low bandwidth and high latency networks. Future work is needed to establish optimal hyper-parameters for NoLoCo and to explore how it can be combined with stale gradient methods or other asynchronous approaches to further reduce the communication overhead.

## 7 REPRODUCIBILITY STATEMENT

All of our experimental results are reproducible. We use only public datasets, models and tokenizers in the paper. We also report all training hyper-parameters and details needed for reproducing the results in Section 4. Source code is also anonymously published in GitHub.

## 8 ETHICAL STATEMENT

We conform to the ICLR code of ethics. Also, we do not make use of LLMs for ideating or writing. LLMs were used for the purposes of this work to train models and evaluate their performance and training time.

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

# A    CONVERGENCE ANALYSIS

## A.1    EXPECTED VALUE AND VARIANCE OF INNER ITERATION

The proposed method in this study aims to lower the communication overhead of data parallel training via the modified Nesterov momentum expression given by Eq. 2. This appendix shows proof that the modified version converges to the vicinity of the real optima when the hyperparameters are chosen appropriately. We structure the proof as follows: this section gives background context and derives expected value and variance for the inner iterations weights; Section A.2 provides convergence proof for the expected value of slow weights; and finally A.3 shows proof for the variance of the slow weights.

To show this we assume a stochastic loss from prior studies Schaul et al. (2013); Wu et al. (2018); Zhang et al. (2019):

$$\mathcal{L}(\theta) = \frac{1}{2}(\theta - c)^{\mathrm{T}} \mathrm{A}(\theta - c), \tag{6}$$

where $c \sim \mathcal{N}(0, \Sigma)$ is a vector valued random variable; A is a positive definite symmetric matrix; and the covariance matrix $\Sigma$ is a constant Zhang et al. (2019). With the above loss function, the minimum value is obtained at $\theta = 0$. Gradient of the loss function is given by:

$$\nabla_\theta \mathcal{L}(\theta) = \mathrm{A}(\theta - c) \sim \mathcal{N}(\mathrm{A}\theta, \mathrm{A}\Sigma\mathrm{A}). \tag{7}$$

For this analysis, we assume that the inner optimizer is using the stochastic gradient descent with a constant learning rate. Stochastic gradient descent updates the (fast) weights according to following rule:

$$\theta_{t,i}^{(k+1)} = (\mathrm{I} - \omega\mathrm{A})\,\theta_{t,i}^{(k)} + \omega\mathrm{A}c, \tag{8}$$

where $(k)$ denotes the (inner) iteration index and $\omega$ is the inner learning rate. With these assumptions, expected value and variance of the fast model weights after one inner iteration are Wu et al. (2018):

$$\mathbb{E}(\theta_{t+1,i}^{(j+1)}) = (\mathrm{I} - \omega\mathrm{A})\,\mathbb{E}(\theta_{t+1,i}^{(j)}), \tag{9}$$

$$\mathbb{V}(\theta_{t+1,i}^{(j+1)}) = (\mathrm{I} - \omega\mathrm{A})\,\mathbb{V}(\theta_{t+1,i}^{(j)})\,(\mathrm{I} - \omega\mathrm{A}) + \omega^2\mathrm{A}\Sigma\mathrm{A}, \tag{10}$$

where I is the identity matrix, and $\theta_{t+1,i}^{(0)} = \phi_{t,i}$. To simplify the notation, we define following shorthands:

$$\mathrm{B} = \mathrm{I} - \omega\mathrm{A}, \tag{11}$$

$$\mathrm{U} = \omega^2\mathrm{A}\Sigma\mathrm{A}. \tag{12}$$

Solving Eqs. 9 and 10 for expected value and variance, we obtain:

$$\mathbb{E}(\theta_{t+1,i}^{(j+1)}) = \mathrm{B}^j\mathbb{E}(\theta_{t+1,i}^{(0)}), \tag{13}$$

$$\mathbb{V}(\theta_{t+1,i}^{(j+1)}) = \mathrm{B}^j\mathbb{V}(\theta_{t+1,i}^{(0)})\mathrm{B}^j + \mathcal{F}^{-1}\left(\mathrm{U} - \mathrm{B}^j\mathrm{U}\mathrm{B}^j\right), \tag{14}$$

where $\mathcal{F}$ is a linear function given by:

$$\mathcal{F}(\mathrm{X}) = \mathrm{X} - \mathrm{B}\mathrm{X}\mathrm{B}. \tag{15}$$

$\mathcal{F}$ is invertible when $\omega > 0$ and $\mathcal{F}(\mathrm{X}) \equiv 0$ when $\omega = 0$.

From Eqs. 13 and 14 we can solve for the outer gradient:

$$
\begin{align}
\mathbb{E}(\Delta_{t,i}) &= \mathbb{E}(\theta_{t+1,i}^{(m)} - \phi_{t,i}) \tag{16} \\
&= (\mathrm{I} - \omega\mathrm{A})^m \, \mathbb{E}(\phi_{t,i}) - \mathbb{E}(\phi_{t,i}) \tag{17} \\
&= ((\mathrm{I} - \omega\mathrm{A})^m - \mathrm{I}) \, \mathbb{E}(\phi_{t,i}), \tag{18}
\end{align}
$$

where $m$ is the number of inner optimizer steps in one outer optimizer step. Similarly for variance,

$$
\begin{align}
\mathbb{V}(\Delta_{t,i}) &= \mathbb{V}\left(\sum_{k=1}^{m} \theta_{t+1,i}^{(k)} - \theta_{t+1,i}^{(k-1)}\right) = \mathbb{V}\left(-\omega \sum_{k=0}^{m-1} \nabla_\theta \mathcal{L}(\theta_{t+1,i}^{(k)})\right) \tag{19} \\
&\approx \omega^2 \sum_{k=0}^{m-1} \mathbb{V}(\nabla_\theta \mathcal{L}(\theta_{t+1,i}^{(k)})) \tag{20} \\
&= \omega^2 \mathrm{A} \sum_{k=0}^{m-1} \left(\mathbb{V}(\theta_{t+1,i}^{(k)}) + \Sigma\right) \mathrm{A} \tag{21} \\
&= \omega^2 \mathrm{A} \left(\sum_{k=0}^{m-1} \mathrm{B}^{k-1} \mathbb{V}(\theta_{t+1,i}^{(0)}) \mathrm{B}^{k-1} + \mathcal{F}^{-1}\left(\mathrm{U} - \mathrm{B}^{k-1}\mathrm{U}\mathrm{B}^{k-1}\right)\right) \mathrm{A} + m\mathrm{U} \tag{22} \\
&= \omega^2 \mathrm{A} \left(\sum_{k=0}^{m-1} \mathrm{B}^{k} \mathbb{V}(\theta_{t+1,i}^{(0)}) \mathrm{B}^{k}\right) \mathrm{A} + \mathrm{R}' \tag{23} \\
&= \omega^2 \mathrm{A} \mathcal{F}^{-1}\left(\mathbb{V}(\theta_{t+1,i}^{(0)}) - \mathrm{B}^{k}\mathbb{V}(\theta_{t+1,i}^{(0)})\mathrm{B}^{k}\right) \mathrm{A} + \mathrm{R}', \tag{24}
\end{align}
$$

where $\mathrm{R}'$ is a constant matrix depending on $\omega$, $\Sigma$, $\mathrm{A}$, and $m$. In Eq. 24 we neglected covariance of non-consecutive fast weights similar to study Zhang et al. (2019). Eq. 24 can be simplified by writing the variance matrix as a vector using following notation:

$$
\mathbb{U}(\theta) \equiv \mathrm{vec}(\mathbb{V}(\theta)) = [\mathbb{V}(\theta)_1, \mathbb{V}(\theta)_2, \cdots], \tag{25}
$$

where $\mathbb{V}(\theta)_1$ and $\mathbb{V}(\theta)_2$ are column vectors of the covariance matrix $\mathbb{V}(\theta)$; and $\mathrm{vec}(\cdot)$ denotes converting a matrix to a vector by concatenating all the column vectors. With this notation we obtain:

$$
\begin{align}
\mathbb{U}(\Delta_{t,i}) &= \mathrm{vec}\left(\omega^2 \mathrm{A} \mathcal{F}^{-1}\left(\mathbb{V}(\theta_{t+1,i}^{(0)}) - \mathrm{B}^{k}\mathbb{V}(\theta_{t+1,i}^{(0)})\mathrm{B}^{k}\right) \mathrm{A} + \mathrm{R}'\right) \tag{26} \\
&= \omega^2 \mathrm{A} \otimes \mathrm{A} \left(\mathrm{I} - \mathrm{B} \otimes \mathrm{B}\right)^{-1} \left(\mathrm{I} - \mathrm{B}^{k} \otimes \mathrm{B}^{k}\right) \mathbb{U}(\theta_{t+1,i}^{(0)}) + \mathrm{vec}(\mathrm{R}'), \tag{27}
\end{align}
$$

where $\otimes$ denotes Kronecker product. We also define:

$$
\mathrm{B}_{\mathbb{V}} = \omega^2 \mathrm{A} \otimes \mathrm{A} \left(\mathrm{I} - \mathrm{B} \otimes \mathrm{B}\right)^{-1} \left(\mathrm{I} - \mathrm{B}^{k} \otimes \mathrm{B}^{k}\right). \tag{28}
$$

## A.2 Expected Value of Outer Iteration

We proceed to show following:

**Theorem 2** *When the outer iteration step count $t \to \infty$, the expected value of the slow weights $\mathbb{E}(\phi_{t,i}) \to 0$.*

We remind the reader that the modified Nesterov momentum used in this study is given by:

$$
\delta_{t,i} = \alpha \delta_{t-1,i} - \frac{\beta}{n}\left(\sum_j \Delta_{t,j}\right) - \gamma\left(\phi_{t,i} - \frac{1}{n}\sum_j \phi_{t,j}\right), \tag{29}
$$

where $n$ is the group size used for the sample means.

The different realizations of the slow weights, $\phi_{t,j}$, are generally not independent due to the path decomposition mechanism and due to the previous outer iterations. However, we will assume that

they would be independent for convergence analysis, which introduces an error in the expected value and variance expressions. This error will become smaller when the data parallel world size becomes larger as the coupling between slow weights becomes weaker.

The optimization method will start from the same instance of slow model weights; hence $\phi_{0,i} \equiv \phi_0$ as the first starting point. All instances of slow weights are updated by the same algorithm and hence the value of $\phi_{t,i}$ depends on two things: (1) what data is used to compute the inner gradients; and (2) what other instances are used in the sample averages. Both of these processes are identically random regarding different instances, and hence we assume that the slow weights have identical distributions. This has following corollaries.

**Lemma 1** *The expected values of the slow weights $\phi_{t,i}$ satisfy:* $\mathbb{E}(\phi_{t,i} - \frac{1}{n}\sum_j \phi_{t,j}) = 0$.

We present a formal proof Lemma 1 in Appendix B. For the variances situation is more complex, and we assume following conjecture based on the above informal reasoning:

**Conjecture 1** *The variances of the slow weights $\phi_{t,i}$ satisfy:* $\mathbb{V}(\phi_{t,i} - \frac{1}{n}\sum_j \phi_{t,j}) \approx 2\left(\frac{n-1}{n}\right)^2 \mathbb{V}(\phi_{t,i})$.

The outer gradients are fundamentally dependent on the current slow weights. We neglect the dependency of the instances outer gradients on other gradients slow weights as before, which will introduce another error in the approximation. This error should diminish as the data parallel world size becomes larger and the coupling between different instances of slow weights becomes weaker. Similar to reasoning with the slow weights, we assume that the outer gradients have identical distributions. With these assumptions, the expected value of $\delta_{t,i}$ becomes:

$$\mathbb{E}(\delta_{t,i}) = \alpha\mathbb{E}(\delta_{t-1,i}) + \beta\mathbb{E}(\Delta_{t,i}), \tag{30}$$

$$= \beta\sum_{k=0}^{t}\alpha^{t-k}\mathbb{E}(\Delta_{k,i}) \tag{31}$$

$$= \beta\sum_{k=0}^{t}\alpha^{t-k}\left(\mathrm{B}^m - \mathrm{I}\right)\mathbb{E}(\phi_{k,i}) \tag{32}$$

that is the same expression as for regular Nesterov momentum. Expected value of the next iteration slow weights becomes:

$$\mathbb{E}(\phi_{t+1,i}) = \mathbb{E}(\phi_{t,i}) + \mathbb{E}(\delta_{t,i}) \tag{33}$$

$$= \mathbb{E}(\phi_{t,i}) + \beta\sum_{k=0}^{t}\alpha^{t-k}\left(\mathrm{B}^m - \mathrm{I}\right)\mathbb{E}(\phi_{k,i}) \tag{34}$$

$$= \mathbb{E}(\phi_{t,i}) + \beta\alpha^t\left(\mathrm{B}^m - \mathrm{I}\right)\sum_{k=0}^{t}\alpha^{-k}\mathbb{E}(\phi_{k,i}) \tag{35}$$

$$= \mathbb{E}(\phi_{t,i}) + \beta\alpha^t\left(\mathrm{B}^m - \mathrm{I}\right)\left(\alpha^{-t}\mathbb{E}(\phi_{t,i}) + \sum_{k=0}^{t-1}\alpha^{-k}\mathbb{E}(\phi_{k,i})\right) \tag{36}$$

$$= \left(\mathrm{I} + \beta(\mathrm{B}^m - \mathrm{I})\right)\mathbb{E}(\phi_{t,i}) \tag{37}$$

$$+ \alpha\left(-\mathbb{E}(\phi_{t-1,i}) + \mathbb{E}(\phi_{t-1,i}) + \beta\alpha^{t-1}\left(\mathrm{B}^m - \mathrm{I}\right)\sum_{k=0}^{t-1}\alpha^{-k}\mathbb{E}(\phi_{k,i})\right) \tag{38}$$

$$= \left(\mathrm{I} + \beta(\mathrm{B}^m - \mathrm{I})\right)\mathbb{E}(\phi_{t,i}) + \alpha\left(\mathbb{E}(\phi_{t,i}) - \mathbb{E}(\phi_{t-1,i})\right) \tag{39}$$

$$= \left(\mathrm{I} + \alpha\mathrm{I} + \beta(\mathrm{B}^m - \mathrm{I})\right)\mathbb{E}(\phi_{t,i}) - \alpha\mathbb{E}(\phi_{t-1,i}) \tag{40}$$

$$= \mathrm{D}\mathbb{E}(\phi_{t,i}) - \alpha\mathbb{E}(\phi_{t-1,i}), \tag{41}$$

where $\mathrm{D} = \mathrm{I} + \alpha\mathrm{I} + \beta(\mathrm{B}^m - \mathrm{I})$. Solving $\mathbb{E}(\phi_{t,i})$ from the above recursive equation using method of characteristics yields:

$$\mathbb{E}(\phi_{t,i}) = \mathrm{C}_1 r_1^t + \mathrm{C}_2 r_2^t, \tag{42}$$

where $C_1$ and $C_2$ are constants independent of $t$; $r_1$ and $r_2$ are the matrix roots of the characteristic polynomial given by:

$$r_1 = \frac{1}{2}\left(D + \sqrt{D^2 - 4\alpha I}\right), \tag{43}$$

$$r_2 = \frac{1}{2}\left(D - \sqrt{D^2 - 4\alpha I}\right). \tag{44}$$

We note that $0 < r_2 \leq r_1 \leq D$ when $0 \leq \alpha < 1$, hence it is sufficient to show that $D^t \to 0$ that happens if and only if all eigen values of $D$, $\mathcal{D}_i$, have less than unit absolute value. Recall, that A is symmetric and positive definite and hence has eigen value decomposition: $A = Q\Lambda Q^T$ where Q is an orthogonal matrix and $\Lambda$ is a diagonal matrix with positive non-zero elements at diagonal. This yields:

$$B^m = \left(I - \omega Q\Lambda Q^T\right)^m \tag{45}$$

$$= \left(Q(I - \omega\Lambda)Q^T\right)^m \tag{46}$$

$$= Q\left(I - \omega\Lambda\right)^m Q^T. \tag{47}$$

Substituting Eq. 47 to definition of D gives:

$$D = I + \alpha I + \beta(B^m - I) \tag{48}$$

$$= I + \alpha I + \beta Q\left((I - \omega\Lambda)^m - I\right)Q^T \tag{49}$$

$$= Q\left(I + (\alpha - \beta)I + \beta(I - \omega\Lambda)^m\right)Q^T, \tag{50}$$

where we can identify that the eigen values are:

$$\mathcal{D}_i = 1 + \alpha - (1 - (1 - \omega\Lambda_i)^m)\beta, \tag{51}$$

where $\Lambda_i > 0$ is the $i^{\text{th}}$ eigen value of A. Convergence of the expected value depends on the hyper-parameters $\alpha$, $\beta$, $\omega$, and $m$. When $m$ is sufficiently large and inner learning rate is chosen to satisfy $0 < \omega\Lambda_i \leq 1$, sufficient condition is $\beta > \alpha$ and $\mathbb{E}(\phi_{t,i}) \to 0$ when $t \to \infty$. Thus the method converges to optimal solution.

### A.3 VARIANCE OF OUTER ITERATION

Finally, we will show following:

**Theorem 3** *When the outer iteration step count $t \to \infty$, the expected value of the slow weights $\mathbb{V}(\phi_{t,i}) \propto \omega^2$.*

The variance of slow weights is given by:

$$\mathbb{V}(\phi_{t+1,i}) = \mathbb{V}(\phi_{t,i}) + \mathbb{V}(\delta_{t,i}) + 2\text{Cov}(\phi_{t,i}, \delta_{t,i}) \tag{52}$$

We only consider direct dependencies of slow weights $\phi_{t,i}$ for the covariance term that becomes:

$$\text{Cov}(\phi_{t,i}, \delta_{t,i}) \approx -\gamma^2 \frac{n-1}{n}\mathbb{V}(\phi_{t,i}), \tag{53}$$

where we omitted the covariance between the outer gradients and corresponding slow weights. The variance of the momentum term becomes:

$$\mathbb{U}(\delta_{t,i}) \approx \alpha^2 \mathbb{U}(\delta_{t-1,i}) + \frac{\beta^2}{n}\mathbb{U}(\Delta_{t,i}) + 2\gamma^2\left(\frac{n-1}{n}\right)^2 \mathbb{U}(\phi_{t,i}) \tag{54}$$

$$= \alpha^2 \mathbb{U}(\delta_{t-1,i}) + \left(\frac{\beta^2 B_{\mathbb{V}}}{n} + 2\gamma^2\left(\frac{n-1}{n}\right)^2\right)\mathbb{U}(\phi_{t,i}) + \frac{\beta^2}{n}\text{vec}(R') \tag{55}$$

$$= \sum_{k=0}^{t-1} \alpha^{2(t-1-k)}\left(\left(\frac{\beta^2 B_{\mathbb{V}}}{n} + 2\gamma^2\left(\frac{n-1}{n}\right)^2\right)\mathbb{U}(\phi_{k,i}) + \frac{\beta^2}{n}\text{vec}(R')\right) \tag{56}$$

$$= \alpha^{2t-2}\sum_{k=0}^{t-1} \alpha^{-2k}\left(C_{\mathbb{V}}\mathbb{U}(\phi_{k,i}) + R''\right), \tag{57}$$

where we assumed that initial momentum is $\mathbb{V}(\delta_{0,i}) \equiv 0$, and used shorthands:

$$C_{\mathbb{V}} = \frac{\beta^2 B_{\mathbb{V}}}{n} + 2\gamma^2 \left(\frac{n-1}{n}\right)^2, \tag{58}$$

$$R'' = \frac{\beta^2}{n} \mathrm{vec}(R'). \tag{59}$$

Substituting Eqs. 53 and 57 into Eq. 52 yields:

$$\mathbb{U}(\phi_{t+1,i}) = \mathbb{U}(\phi_{t,i}) - 2\gamma^2 \frac{n-1}{n} \mathbb{U}(\phi_{t,i}) + \alpha^{2t-2} \sum_{k=0}^{t-1} \alpha^{-2k}(C_{\mathbb{V}}\mathbb{U}(\phi_{k,i}) + R'') \tag{60}$$

$$= \left(1 - 2\gamma^2 \frac{n-1}{n}\right)\mathbb{U}(\phi_{t,i}) + \alpha^{2t-2} \sum_{k=0}^{t-1} \alpha^{-2k}(C_{\mathbb{V}}\mathbb{U}(\phi_{k,i}) + R''). \tag{61}$$

The last sum-term can be rearranged as follows:

$$\alpha^{2t-2} \sum_{k=0}^{t-1} \alpha^{-2k}(C_{\mathbb{V}}\mathbb{U}(\phi_{k,i}) + R'') \tag{62}$$

$$= C_{\mathbb{V}}\mathbb{U}(\phi_{t-1,i}) + R'' + \alpha^2 \left(\alpha^{2(t-2)} \sum_{k=0}^{t-2} \alpha^{-2k}(C_{\mathbb{V}}\mathbb{U}(\phi_{k,i}) + R'')\right) \tag{63}$$

$$= C_{\mathbb{V}}\mathbb{U}(\phi_{t-1,i}) + R'' + \alpha^2 \left(\mathbb{U}(\phi_{t,i}) - \left(1 - 2\gamma^2 \frac{n-1}{n}\right)\mathbb{U}(\phi_{t-1,i})\right) \tag{64}$$

$$= \alpha^2 \mathbb{U}(\phi_{t,i}) + \left(C_{\mathbb{V}} - \alpha^2 \left(1 - 2\gamma^2 \frac{n-1}{n}\right)I\right)\mathbb{U}(\phi_{t-1,i}) + R''. \tag{65}$$

Substituting Eq. 65 to Eq. 61 gives:

$$\mathbb{U}(\phi_{t+1,i}) = d_{\mathbb{V}}\mathbb{U}(\phi_{t,i}) + E_{\mathbb{V}}\mathbb{U}(\phi_{t-1,i}) + R'', \tag{66}$$

where $d_{\mathbb{V}}$ and $E_{\mathbb{U}}$ are given by:

$$d_{\mathbb{V}} = 1 + \alpha^2 - 2\gamma^2 \frac{n-1}{n}, \tag{67}$$

$$E_{\mathbb{V}} = C_{\mathbb{V}} - \alpha^2 \left(1 - 2\gamma^2 \frac{n-1}{n}\right)I. \tag{68}$$

Solving $\mathbb{U}(\phi_{t,i})$ by method of characteristics gives following solution:

$$\mathbb{U}(\phi_{t,i}) = C_1' v_1^t + C_2' v_2^t + R'', \tag{69}$$

where $C_1'$ and $C_2'$ are constants independent of $t$; $v_1$ and $v_2$ are the matrix roots of the characteristic polynomial given by:

$$v_1 = \frac{1}{2}\left(d_{\mathbb{V}}I + \sqrt{d_{\mathbb{V}}^2 I - 4E_{\mathbb{V}}}\right), \tag{70}$$

$$v_2 = \frac{1}{2}\left(d_{\mathbb{V}}I - \sqrt{d_{\mathbb{V}}^2 I - 4E_{\mathbb{V}}}\right). \tag{71}$$

For real roots, we have following bounds: $0 \leq \|v_2\|_2 \leq \|v_1\|_2 \leq |d_{\mathbb{V}}|$. For the variance to remain bounded as $t \to \infty$ we must have $|d_{\mathbb{V}}| < 1$. Solving for $\gamma$ we arrive at condition:

$$\sqrt{\frac{n}{2(n-1)}}\alpha < \gamma < \sqrt{\frac{n}{2(n-1)}(2+\alpha^2)}. \tag{72}$$

When hyper-parameters satisfy Eq. 72, $\mathbb{U}(\phi_{t,i}) \to R''$ when $t \to \infty$. The leading order term of $R''$ in terms of inner learning rate is $\propto \omega^2$. Hence, the variance $\|\mathbb{U}(\phi_{t,i})\|_2 \propto \omega^2$ when $\omega \to 0$ which proofs that the method converges to the correct optima and the variance of the optima is proportional to square of inner learning rate.

# B PROOF FOR IDENTICAL MODEL EXPECTED VALUES

Reminder that after $k + 1$ steps, the expected fast weights are:

$$\mathbb{E}[\theta_{t+1,i}^{k+1}] = \mathrm{B}^k \mathbb{E}[\phi_{t,i}], \tag{73}$$

where $\mathrm{B} = \mathrm{I} - \omega \mathrm{A}$. We prove that for any arbitrary step $t$ the expected difference between an arbitrary $\phi_{t,i}$ and the average of several other $\frac{1}{n} \sum_j^n \phi_{t,j}$ is 0:

$$\mathbb{E}[\phi_{t,i} - \frac{1}{n} \sum_j^n \phi_{t,j}] = \mathbb{E}[\phi_{t,i}] - \frac{1}{n} \sum_j^n \mathbb{E}[\phi_{t,j}] = 0. \tag{74}$$

Reminder about some of the variables:

$$\Delta_{t,i} = \theta_{t+1,i}^{k+1} - \phi_{t,i} \tag{75}$$

$$\phi_{t+1,i} = \phi_{t,i} + \delta_{t,i} \tag{76}$$

$$\delta_{t,i} = \alpha \delta_{t-1,i} - \frac{\beta}{n} \sum_j^n \Delta_{t,j} - \gamma \left( \phi_{t,i} - \frac{1}{n} \sum_j^n \phi_{t,j} \right). \tag{77}$$

It is trivial to see that at step 1 the property holds:

$$\phi_{0,i} = \phi_0 \tag{78}$$

$$\delta_{0,i} = 0 \tag{79}$$

$$\delta_{1,i} = -\frac{\beta}{n} \sum_j^n \left( \theta_{t+1,i}^{k+1} - \phi_0 \right) \mathbb{E}[\delta_{1,i}] \tag{80}$$

$$= -\frac{\beta}{n} \sum_j^n \left( \mathrm{B}^k \mathbb{E}[\phi_0] - \mathbb{E}[\phi_0] \right) \tag{81}$$

$$= -\beta (\mathrm{B}^k - \mathrm{I}) \mathbb{E}[\phi_0], \text{ and} \tag{82}$$

$$\mathbb{E}[\phi_{1,i}] = \mathbb{E}[\phi_0] - \beta (\mathrm{B}^k - \mathrm{I}) \mathbb{E}[\phi_0]. \tag{83}$$

Since the value does not depend on $i$, then the value will be identical across all replicas, thus $\mathbb{E}[\phi_{1,i}] - \frac{1}{n} \sum_j^n \mathbb{E}[\phi_{1,j}] = 0$.

We proceed next to show that if the property holds for an arbitrary step $T$ and for all steps prior to that $t \in [0, T]$, then it also holds for $T + 1$.

At step $T + 1$ we have:

$$\delta_{T+1,i} = \alpha \delta_{t,i} - \frac{\beta}{n} \sum_j^n \Delta_{T,j} - \gamma \left( \phi_{T,i} - \frac{1}{n} \sum_j^n \phi_{T,j} \right). \tag{84}$$

Taking expected value of Eq. 84 yield:

$$\mathbb{E}[\delta_{T+1,i}] = \alpha \mathbb{E}[\delta_{T,i}] - \frac{\beta}{n} \sum_j^n \mathbb{E}[\Delta_{T,j}] - \gamma \left( \mathbb{E}[\phi_{T,i}] - \frac{1}{n} \sum_j^n \mathbb{E}[\phi_{T,j}] \right). \tag{85}$$

The right term reduces to 0 and thus we are left with:

$$\mathbb{E}[\delta_{T+1,i}] = \alpha \mathbb{E}[\delta_{T,i}] - \frac{\beta}{n} (\mathrm{B}^k - \mathrm{I}) \sum_j^n \mathbb{E}[\phi_T] \tag{86}$$

$$= \alpha \mathbb{E}[\delta_{T,i}] - \beta (\mathrm{B}^k - \mathrm{I}) \mathbb{E}[\phi_T], \tag{87}$$

| Method | 1M | 2M |
|--------|------|------|
| FSDP | 19.6 | 18.0 |
| DiLoCo | 21.0 | 19.7 |
| NoLoCo | 20.9 | 19.3 |

Table 4: Summary of final perplexity numbers from Reddit data with varying global batch size. All results are from the medium model size, 64 accelerators, four pipeline stages, and data parallel world size of sixteen.

and

$$\mathbb{E}[\phi_{T+1,i}] = \mathbb{E}[\phi_{T,i}] + \alpha\mathbb{E}[\delta_{t,i}] - \beta(\mathrm{B}^k - \mathrm{I})\mathbb{E}[\phi_T]. \tag{88}$$

Finally,

$$\mathbb{E}[\phi_{T+1,i}] - \frac{1}{n}\sum_j^n \mathbb{E}[\phi_{T+1,j}] = \mathbb{E}[\phi_{T,i}] + \alpha\mathbb{E}[\delta_{T,i}] - \beta(\mathrm{B}^k - \mathrm{I})\mathbb{E}[\phi_T] \tag{89}$$

$$- \frac{1}{n}\sum_j^n \left(\mathbb{E}[\phi_{T,j}] + \alpha\mathbb{E}[\delta_{T,j}] - \beta(\mathrm{B}^k - \mathrm{I})\mathbb{E}[\phi_T]\right) \tag{90}$$

$$= \alpha\mathbb{E}[\delta_{T,i}] - \frac{1}{n}\sum_j^n \left(\alpha\mathbb{E}[\delta_{T,j}]\right) \tag{91}$$

$$= \mathbb{E}[\phi_{T-1,i}] - \frac{1}{n}\sum_j^n \left(\mathbb{E}[\phi_{T-1,j}]\right) \tag{92}$$

$$- \mathbb{E}[\phi_{T,i}] + \frac{1}{n}\sum_j^n \left(\mathbb{E}[\phi_{T-1,j}]\right) = 0. \tag{93}$$

Thus the property holds for step $T + 1$, and by proof by induction it holds for all $T$ since the first step holds.

## C  HYPER-PARAMETER ABLATIONS

DiLoCo and NoLoCo are sensitive to the used batch size. Table 4 present how increasing the batch size affects the results. Increasing the batch size will also increase the number of tokens models observe during training as well as linearly increase the training cost.

## D  TRAINING LOSSES

Figure 6 shows training data cross-entropy for all methods and model sizes on Reddit data. Training cross-entropy is averaged over consecutive 200 (inner) training steps to reduce noise in the data.

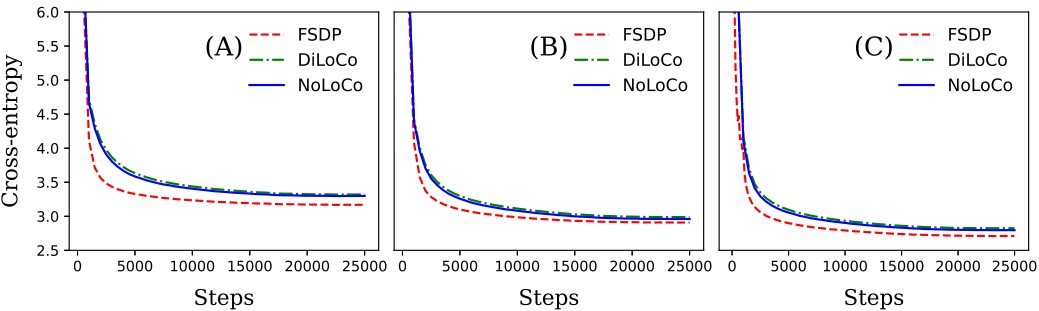

Figure 6: Reddit training cross-entropy of different training methods at different optimizer step counts. Solid blue curve is NoLoCo, dashed red curve show FSDP, and dashed green curve is DiLoCo. (A): Small model with data parallel world size of $8$ and two pipeline stages; (B): Medium model with data parallel world size of $8$ and two pipeline stages. (C): Large model with data parallel world size of $16$ and four pipeline stages.

