# OpenReview forum: "NoLoCo: No-all-reduce Low Communication Training Method for Large Language Models"
_ICLR.cc/2026/Conference — Submitted to ICLR 2026_

### Official Review · Reviewer_KLKc · 2025-10-27

**Soundness:** 2
**Presentation:** 3
**Contribution:** 2
**Rating:** 2
**Confidence:** 3

**Summary:**

This paper try to eliminating global collective communication to reduce the e2e training communication overhead. This can be useful in extreme cases where cross-node network bandwidth is limited.

The authors proposed NoLoCo method, which does dynamic routing on PP dimension (similar as dynamic routing in MoE expert tokens), but do need model synchronization on DP dimension. However, loss divergence issue has been observed.

**Strengths:**

1. The proposed idea is interesting (i.e. doing dynamic routing in PP stages). Also picking PP is practical since it has minimum communication volumes compared to other parallelism strategies.

2. A detailed convergence analysis in appendix is a plus.

**Weaknesses:**

1. In general, eliminating global communication will lead to model divergence. In practice, training a model has much worse model serving accuracy, this training itself will be a complete waste of compute resources. Therefore, only try to optimizing/eliminating communcation overhead is not a good strategy.

2. For experiments, the paper does not mention any details on which LLama vesion is used? e.g. LLama2? llama3? which can be a huge difference on perplexity or loss.

3. the results of Figure 2, it shows there is noticeable and significant loss/PPL gaps between FSDP and proposed NoLoCo strategy, which makes the proposed method not practical in real model training scenarios.

4. Although the paper has some "proof" of model convergence with dynamic PP routing, as shown in later result (e.g. Table2 and Figure 2), the model divergence issue is quite severe. Therefore, it makes the whole NoLoCo scheme less practical, mainly because the loss divergence issue.

**Questions:**

1. In Figure 2, why larger models and more PP stages will make PPL differnce less compared wth FSDP?

2. In practice, nowadays it is very rare to see communication bandwidth is less than 1GB/s. So the Table 3 comparison number on 100Mb/s ( 12.5MB/s) and 1Gb/s (125MB/s) are less practical and less convincing.

3. Theoretically, doing dynamic routing on PP stages will definitely cause loss divergence issue. The high level idea is, difference PP stages of a single PP group are training on different input data at the same iteration, this will definitely introduce errors and wrong momentum.

---

> ### Author Response · Authors · 2025-11-20
> **Response to Reviewer Comments**
>
> We would like to thank the reviewer for their comments and issues raised, as they help us clarify the ideas in our paper. Below we address the issues raised by the reviewer.
>
> **Response to Weakness 1** We agree with the reviewer that only eliminating the communication is not a good strategy; for that reason, we proposed our modified Nesterov momentum, which reduces the divergence of individual models.
>
> We have provided theoretical proof for model convergence and empirically demonstrated that the model weights do not diverge for multiple datasets and model sizes.
>
> **Comment:** *Which Llama vesion is used?*
>
> **Response:** We used the Llama3 base class from HuggingFace with some modifications regarding parallelization strategy. Details are available in the manuscript GitHub repository. However, all the models are trained from scratch, thus the model cannot be referred as Llama2 nor Llama3 directly as it is not derived from any of the official checkpoints. We will clarify this point in the revised manuscript.
>
> **Comment:** *In Figure 2, there is noticeable loss/PPL gap between FSDP and NoLoCo, which makes it not practical in real model training scenarios.*
>
> **Response:** We would like to point out a few important details regarding Fig. 2: (i) NoLoCo is better than FSDP at any point of time in terms of wall-time (with moderate interlink speeds). (ii) The final PPL convergence is impacted by learning rate decay that we used and increasing the number of tokens (with lower decay rate) changes the final result. FSDP is also prohibitively slow with low interlink speeds and for instance repeating the 7B training run from the paper would take a year.
>
> For these reasons, in the case of distributed training with low/medium connection speeds, NoLoCo converges much faster than FSDP in wall-clock time.
>
> **Comment:** *Although the paper has some "proof" of model convergence with dynamic PP routing, as shown in later result, the model divergence issue is quite severe.*
>
> **Response:** We do not have any results in the paper that shows any divergence of the models regarding training loss nor any claims of that nature. If the reviewer is referring to the difference between perplexity of NoLoCo and FSDP, NoLoCo is designed for low bandwidth networks and aims to reduce the communication overhead while having minimal effect on convergence.
>
> Compared to the baseline DiLoCo, NoLoCo has better perplexity results, while also requiring less communication. This is because of our modified Nesterov optimizer. Compared with FSDP, NoLoCo has significantly better wall-clock time performance in the low/medium bandwidth networks. It has been known, and also confirmed with our experiments, that FSDP is only feasible in high-speed connections in data centers because of the heavy communication requirement.
>
> **Response to Question 1** We believe that the hyper-parameters are more suitable for the decentralized methods with larger models, and as a result produce a smaller gap compared with the FSDP on training step basis. PP stages may have a minor effect in NoLoCo as they promote better weight similarity across different model instances which can have a small positive effect.
>
> **Response to Question 2** We believe the reviewer is referring to interlinks inside data centers that indeed are typically faster than 1Gb/s and dedicated machine learning clusters can have interlink speeds exceeding 1Tb/s. However, our work is targeting training using large number of consumer grade machines which have much lower interlink speeds. The average consumer-grade internet download speed ranges between 10-400 Mb/s depending on the country [1], and the upload speed is generally much lower. Therefore, the comparison in Table 3 is highly relevant for the intended application.
>
> **Response to Question 3** We have two momentum terms in our optimizer: (i) one used for the inner optimizer inside Adam, and (ii) the outer momentum used in our modified Nesterov optimizer. For the inner optimizer, the momentum term is an ensemble over different pipeline stages that it interacts with and will likely have a higher variance compared with traditional training. However, this does not necessarily lead it to diverge and we did not observe any divergence of loss in any of our experiments. The outer optimizer momentum is an ensemble over all the different model instances and in this case having dynamic PP routing actually reduces the variance as shown in Figure 5. Finally, our perplexity results  are slightly worse than FSDP with respect to the number of iterations, which is caused by much fewer sycnhronisations. However, as mentioned above, because of the huge throughput advantage compared to FSDP (given in Table 3), NoLoCo converges much faster than FSDP.
>
> [1] Speedtest.net, 'Median Country Speeds (Updated October 2025)', https://www.speedtest.net/global-index

---

### Official Review · Reviewer_fHc3 · 2025-11-01

**Soundness:** 3
**Presentation:** 3
**Contribution:** 3
**Rating:** 6
**Confidence:** 4

**Summary:**

This paper proposes NoLoCo, a decentralized training method for large language models that eliminates global all-reduce synchronization. Building on DiLoCo, which reduces communication by performing infrequent global synchronizations between local training phases, NoLoCo further minimizes overhead by synchronizing only random pairs of workers using a modified Nesterov momentum optimizer that maintains model stability. Combined with dynamic pipeline routing, this approach enables efficient large-scale training on low-bandwidth or high-latency networks. Experiments on different sizes of LLaMA models show that NoLoCo achieves up to 4% faster convergence than DiLoCo while greatly reducing communication cost.

**Strengths:**

•	The key innovation of this work is to remove global all-reduce synchronization by introducing local, pairwise averaging.
•	The authors propose a new variant of Nesterov momentum with an additional local averaging term to prevent divergence, supported by theoretical convergence analysis.
•	The authors provide mathematical proofs showing convergence and variance behavior, linking stability to the inner learning rate.

**Weaknesses:**

•	Only empirically compare NoLoCo against the original DiLoCo baseline, despite acknowledging multiple improved variants.
•	Convergence proof assumes independence between replicas and IID data, which may not hold in real heterogeneous clusters.
•	Largely reused learning rates and batch sizes from prior FSDP studies and fixed outer-loop settings without systematic or sensitivity analysis.
•	The authors may want to separately test pairwise averaging, modified Nesterov, and random routing to show each component’s contribution.

**Questions:**

•	The paper references multiple improved versions but compares only with the original DiLoCo; including these baselines would better show whether NoLoCo truly advances beyond recent low-communication optimizers. If cannot make the comparison, the authors may want to provide a detailed explanation of why these variants were excluded and how NoLoCo differs from them in design or communication efficiency.
•	The paper integrates three innovations: pairwise averaging, modified Nesterov momentum, and random pipeline routing. However, do not disentangle their effects. The authors may want to conduct ablation studies that isolate each component if possible to clarify which element primarily drives convergence speed, stability, and communication efficiency.
•	NoLoCo introduces several new hyperparameters (like outer momentum, outer learning rate, local averaging strength, subgroup size, and outer step frequency), but the paper provides little analysis of how these parameters influence stability, convergence, or communication cost. Could the authors discuss which hyperparameters are most critical, how sensitive the method is to their choice, and how much tuning effort practitioners should expect compared to standard distributed training?

---

> ### Author Response · Authors · 2025-11-20
> **Responses to Reviewer Comments**
>
> We would like to thank the reviewer for their comments and issues raised, as they help us clarify the ideas in our paper. Below we address the issues raised by the reviewer.
>
> **Comment:** *Only comparison with DiLoCo, not variants*
>
> **Response:** Our intention in comparing with DiLoCo is to show how the modified Nesterov optimizer compares with the regular Nesterov optimizer in a DiLoCo-style setting. Recent improvements to DiLoCo come from various asynchronous gradient updates that allow overlapping the communication with the computation. Our research direction is complementary to this by aiming to reduce the actual amount of data communicated.
>
> It should also be noted that model parallelism will require some communication bandwidth which will compete with the gradient communication for the bandwidth. Some communication libraries (i.e. torch distributed) do not allow overlapping communication. Asynchronous methods such as streaming DiLoCo will lose a lot of the benefits when such backend is combined with model parallelism.
>
> **Comment:** *Convergence proof assumes independence between replicas and IID data*
>
> **Response:** Since our work uses the same data assumptions as DP and its extensions like DiLoCo (unlike a federated learning-like setting), we assume IID data.
> Also, at the present time it is unclear how the replicas depend on each other and this would need to be modeled before the system can be analyzed further.
>
>
> **Comment:** *Ablation studies on pairwise averaging, modified Nesterov momentum, and random pipeline routing.*
>
> **Response:** Pairwise averaging is the central part of the modified Nesterov optimizer and cannot be studied in isolation. Using the full data parallel world to perform the averaging will collapse the modified Nesterov back to the regular Nesterov optimizer.
>
> Random routing is only applicable when there is more than one pipeline stage. We show results without the pipeline routing (PP=1) for the small and medium size models. Results without pipeline parallelism are used to probe the effect of the modified Nesterov optimizer in isolation (DiLoCo uses the regular Nesterov optimizer). We were unable to do this for the large model due to limited GPU memory.
>
> **Comment:** *Re-used LR and batch sizes from prior studies*
>
> **Response:** Indeed, we intended to use prior learning rates and batch sizes to give a fair comparison between different methods. However, we agree that they may not be ideal for the decentralized methods.
>
> **Comment:** *Hyperparameter selection and their influence*
>
> **Response:** We agree with the reviewer that analyzing these new hyperparameters is important. We provided some ablation studies for local group size (using 2 and all the data parallel workers), and batch size but we agree with the reviewer that exploring the gamma-parameter more in depth is valuable.
>
> We hope that our answers address the comments of the reviewer. We are happy to answer any further comments the reviewer might have.

---

### Official Review · Reviewer_2LMt · 2025-11-06

**Soundness:** 2
**Presentation:** 2
**Contribution:** 3
**Rating:** 4
**Confidence:** 3

**Summary:**

This paper proposes NoLoCo, a novel optimization method designed to eliminate the need for collective communication during distributed training of large language models (LLMs). Instead of explicitly synchronizing model parameters, NoLoCo achieves implicit synchronization through a variant of the Nesterov momentum optimizer. Specifically, model weights are partially averaged with those of a randomly selected peer, allowing for communication-efficient and scalable model training. The authors provide both theoretical convergence guarantees for the proposed algorithm and empirical experiments demonstrating its efficiency in large-scale language model training.

While the idea is innovative and the theoretical aspect is solid, the experimental studies are incomplete, especially regarding the ablation and sensitivity analyses needed to confirm the importance of the paper’s key design choices. I will raise my score to positive if the authors address my concerns.

**Strengths:**

1. The paper provides both theoretical and empirical validation for the proposed optimization method, which strengthens its overall contribution and credibility.
2. NoLoCo effectively reduces communication overhead compared to state-of-the-art methods, particularly DiLoCo, while achieving faster convergence in language model training.
3. The idea of replacing global synchronization with randomized local interactions is novel and practically meaningful, offering a potential improvement for large-scale distributed training systems.

**Weaknesses:**

1. Ablation studies are insufficient. The paper lacks experiments isolating the impact of the “outer optimizer step with modified Nesterov momentum,” which is one of the paper’s core contributions. Specifically, it is unclear how performance would change if the original (unmodified) Nesterov momentum were used instead of the modified version described in Equation (2). Without further ablation to isolate the effect of the modified Nesterov momentum, it remains unclear whether this component is indeed critical to NoLoCo’s improved performance.
2. The paper omits ablations on critical hyperparameters such as:
   - The group size (n) in section 3.2, which likely influences both communication cost and model performance (e.g., perplexity).
   - The number of inner optimizer steps, which differs between NoLoCo (50) and DiLoCo (100) in the main comparisons, but the rationale for this choice is not adequately explained.
     Including these analyses would provide deeper insight into NoLoCo’s behavior and fairness in comparison.
3. The implementation details in Section 3 are not sufficiently clear, making it challenging to fully grasp how NoLoCo is practically realized.

**Questions:**

1. What would happen if the modified Nesterov momentum (Equation 2) were replaced with the original Nesterov momentum? How does this change impact convergence and model performance?
2. How does group size (n) affect communication efficiency and model perplexity? Can the authors provide an ablation study?
3. How does varying the number of inner optimizer steps influence both communication cost and performance?
4. During the outer optimizer step, are the local groups fixed or re-sampled randomly at each round?
5. Since each step only performs partial synchronization within groups, is there a final global synchronization among all subgroups at the end of training? If not, wouldn’t this result in $\frac{N}{n}$ slightly different model replicas?
6. In Figure 4, the variable *n* is annotated as “world size,” but in Section 3.2 it denotes “group size.” Are these the same variable or different? The notation may be confusing.

---

> ### Author Response · Authors · 2025-11-20
> **Responses to Reviewer Comments**
>
> We would like to thank the reviewer for their comments and issues raised, as they help us clarify the ideas in our paper. Below we address the issues raised by the reviewer.
>
> **Comment:** *Insufficient ablation study on comparison with unmodified Nesterov*
>
> **Response:** We compared our work directly to DiLoCo, which is identical to using unmodified Nesterov momentum.
>
> **Comment:** *Missing ablation results on critical hyperparameters*
>
> **Response:** We used the smallest group size (two members) to showcase the idea. Due to the computational cost of the experiments, we were unable to explore the effect of larger group sizes systematically. We believe that the behavior will approach the one in DiLoCo as our algorithm becomes identical to DiLoCo in the extreme where the group size is identical to the world size. Our choice of using 50 steps for NoLoCo was motivated by the fact that the data transferred will still be less than in DiLoCo (when using tree reduce) and we avoid the global barrier. In tree-reduce, children send the weights to the parent that computes the reduction, and sends the result to its parent. Once the root of the tree has received the final result, it will broadcast the result down the tree. Pairwise reduction can be thought as a single step of tree-reduce at the leaf level without the need for the interior nodes. This choice is aimed to portray both methods evenly. We leave extensive ablation studies as future work.
>
>
> **Comment:** *Unclear implementation details*
>
> **Response:**
> Due to page limitations, some experimental details are given in Appendices C and D.
>
> We have supplemented the manuscript with a GitHub repository that shows the full implementation with all the details for interested readers. We can provide any additional details if the reviewer feels any particular aspect of the algorithm should be covered in greater detail in the manuscript.
>
> **Comment:** *Convergence effect of the modified Nesterov momentum (Equation 2) with the original Nesterov momentum*
>
> **Response:** With the original Nesterov momentum, the training algorithm becomes identical to DiLoCo that has been published before. We provide comparison with DiLoCo for all experiments.
>
> **Comment:** *The effect of group size (n) on communication efficiency and model perplexity*
>
> **Response:** We provide two data points in the paper for the extremes: NoLoCo with group size of 2 and DiLoCo (group size is the same as the data parallel world size). Both methods are fairly close to each other and we hypothesize that the intermediate values would fall between the two extremes.
>
> **Comment:** *The effect of the number of inner optimizer steps on communication cost and performance*
>
> **Response:** The number of inner optimizer steps influences directly how often the outer step communication is performed. That has a linear effect on the outer all-reduces which are communication intensive. The effect of inner optimizer steps on accuracy is more complex, considering the computational costs, we leave this as future work.
>
> **Comment:** *Are the local groups fixed or re-sampled at each round?*
>
> **Response:** We re-sample the groups at every step randomly. We will clarify this detail in the final version of the manuscript.
>
> **Comment:** *Is there a final global synchronization? or the final models are different?*
>
> **Response:** Indeed, the reviewer is correct in pointing out that each replica will have a small difference in value that is controlled by the smallest learning rate.
> We did not perform any final synchronization, but the model replicas will be arbitrarily close to each other when the learning rate decay is applied during training.
>
> **Comment:** *Re-use of notation $n$.*
>
> **Response:** We thank the reviewer for pointing out the re-use of the same notation $n$ for both world size and group size.
> These are meant to be different variables and we will resolve this confusion for the revised manuscript. Figure 4 "n" refers to world size, but the "n" used as group size. We will use a different symbol for the Figure 4 to avoid confusion pointed out by the reviewer.
>
> We hope that our answers address the comments of the reviewer. We are happy to answer any further comments the reviewer might have.

---

> > ### Comment · Reviewer_2LMt · 2025-11-23
> > **Response to Rebuttal by Reviewer 2LMt**
> >
> > I thank the authors for their detailed response and for clarifying the notation issue regarding $ n$. However, after carefully reading the rebuttal, I still have three major concerns regarding the comparison with DiLoCo, the group size assumption, and the final model convergence.
> >
> > **1. Clarification on the "Identity" to DiLoCo (Section 3.1 vs 3.2)**
> > The authors state in the rebuttal: *"With the original Nesterov momentum, the training algorithm becomes identical to DiLoCo."*
> > However, based on Section 3 of the paper, NoLoCo consists of two distinct components:
> > 1.  **Inner Optimizer Step via Dynamic Pipeline Routing** (Section 3.1)
> > 2.  **Outer Optimizer Step with Modified Nesterov Momentum** (Section 3.2)
> >
> > My understanding is that DiLoCo does not utilize the "Dynamic Pipeline Routing" described in Section 3.1. Therefore, even if one replaces the *Modified* Nesterov Momentum (Eq 2) with the *Original* Nesterov Momentum in the outer loop, the presence of the Dynamic Pipeline Routing (Section 3.1) in the inner loop should still make the method distinct from standard DiLoCo.
> >
> > Could the authors please clarify this? My request for an ablation study was specifically to isolate the contribution of Section 3.2. I would like to see the performance of **[NoLoCo with Dynamic Pipeline Routing + Original Nesterov Momentum]**. This is critical to understand if the gain comes from the routing strategy or the modified momentum.
> >
> > **2. Insufficient Data Points for Group Size ($n$)**
> > Regarding the group size, the authors state: *"We hypothesize that the intermediate values would fall between the two extremes."*
> > While I understand the computational constraints, relying on a hypothesis based on only two extreme data points ($n=2$ and $n=\text{World Size}$) is scientifically insufficient to establish a trend. The relationship might not be linear or monotonic. To validate the robustness of NoLoCo, it is necessary to observe at least one intermediate value (e.g., $n=4$ or $n=8$) to confirm that the performance degradation is indeed gradual and predictable, rather than suffering a sharp drop-off as soon as $n > 2$.
> >
> > **3. Theoretical Basis for Replica Convergence**
> > The authors state: *"The model replicas will be arbitrarily close to each other when the learning rate decay is applied during training."*
> > This is a strong claim and is not immediately obvious. While learning rate decay reduces the step size, it does not mathematically guarantee that distinct subgroups (which never synchronize globally) will converge to the *same* point in a non-convex landscape; they might converge to different local minima "arbitrarily close" in loss value, but not necessarily in parameter space.
> > Is there a theoretical proof provided in the appendix that guarantees the parameters of different groups converge to the same values? Alternatively, did the authors measure the Euclidean distance or Cosine similarity between the replicas at the end of training to empirically support this claim?
> >
> > I remain open to raising my score, but I need these specific clarifications to be confident in the method's soundness.

---

> ### Author Response · Authors · 2025-11-23
> **Response to Comments**
>
> Authors like to thank reviewer for their swift response.
>
> **Response to Clarification on the "Identity" to DiLoCo (Section 3.1 vs 3.2)**
>
> Authors wanted to clarify that when PP=1, there is no dynamic pipeline routing and the difference with DiLoCo and NoLoCo is only modified Nesterov momentum equation. Otherwise, reviewer is absolutely correct that there is two differences between DiLoCo and NoLoCo.
>
> **Response to Insufficient Data Points for Group Size ($n$)**
>
> Authors agree that tuning the hyper-parameter $n$ may be important, and that it could behave in non-linear manner. However, we show extensive evidence that the method works for pairwise interactions under wide range of conditions. We also supplement the experimental evidence with an analytical proof that is valid for all values of $n$. Authors feel that the case $n>2$ can be seen as generalization of the current work, and hence should not be grounds for rejection. We are happy to make this limitation more transparent in the revised manuscript.
>
> **Response to Theoretical Basis for Replica Convergence**
>
> Authors agree with reviewer that for non-convex case the replicas could potentially get stuck in different local optima - especially if the hyper-parameter $\gamma$ is very small such that the averaging component is very small compared to gradients. We did not observe this in any of our experiments, but this remains a theoretical possibility nevertheless. Authors are happy to discuss this failure mode in the revised manuscript. The standard deviation shown in Fig 3B is essentially Euclidian distance between the model weights as variance is the sum of the square differences.

---

> > ### Comment · Reviewer_2LMt · 2025-11-27
> >
> > I thank the authors for their response.
> >
> > Regarding **Point 1 (Identity to DiLoCo)** and **Point 3 (Replica Convergence)**, I am satisfied with the clarifications provided.
> >
> > However, regarding **Point 2 (Insufficient Data Points for Group Size $n$)**, I am **not convinced** by the rebuttal.
> >
> > The authors argue that "extensive evidence that the method works for pairwise interactions" combined with analytical proofs is sufficient. I respectfully disagree.
> > The pairwise case ($n=2$) represents the minimal complexity for this method. As the group size increases to $n > 2$, the intra-group communication complexity increases (potentially scaling up to $O(n^2)$ depending on the topology/implementation), and the synchronization dynamics change. Demonstrating success at $n=2$ does not automatically guarantee that the method remains efficient or effective at larger group sizes.
> >
> > Without empirical evidence for an intermediate value (e.g., $n=4$ or $n=8$), it is difficult to verify the method's generalization ability and scalability. Therefore, I maintain that providing experimental results for $n > 2$ is necessary to substantiate the claims made in the paper.

---

> > > ### Author Response · Authors · 2025-11-30
> > > **Response to Reviewer 2LMt**
> > >
> > > We thank reviewer for their comments but authors wanted to clarify few possible misconceptions. We are not making any claims for larger averaging groups sizes ($n>2$) in the current manuscript with an exception of the proof included in the Appendix. Hence, we believe that this is not needed to *substantiate the claims made in the paper* as we are not making any claims on effects of using larger averaging group. Regarding scalability we show that using pairwise averaging works for all world sizes considered in the paper, and authors respectfully disagree that analyzing the more complex groups is necessary to show scalability.

---

### Meta-Review · Area_Chair_juKU · 2026-01-06

**Summary:**

The paper proposes a combination of randomized dynamic pipeline routing and gossip averaging. Experiments on language modelling are demonstrated in low bandwith training scenarios. Reviewers expressed concerns about a lack of compared methods, ablations, and assumptions on the theory. The meta-reviewer notes an additional concern in the paper. The related work does not adequately situate NoLoCo within the well-established literature on decentralized and gossip-based optimization. While DiLoCo is an important recent baseline, the core mechanism of NoLoCo — pairwise local averaging combined with local updates  appears to be a gossip-style decentralized SGD method with local updates. There exists a large body of work (see examples below) on this including ones that study  momentum methods that are not discussed. These works and many followups directly study similar problems to the proposed algorithm and propose extensive theory all of which is not situated or compared in this work.

Lian et al. Can Decentralized Algorithms Outperform Centralized Algorithms? A Case Study for Decentralized Parallel Stochastic Gradient Descent

Lian et al. Asynchronous Decentralized Parallel Stochastic Gradient Descent

Koloskova, Stich & Jaggi  Decentralized Stochastic Optimization and Gossip Algorithms with Compressed Communication

Koloskova et al. A Unified Theory of Decentralized SGD with Changing Topology and Local Updates

Assran et al. Stochastic Gradient Push for Distributed Deep Learning

Wang et al SlowMo: Improving Communication-Efficient Distributed SGD with Slow Momentum

**Reviewer Concerns:**

- Ablations partially addressed but more pending (e.g. gamma, more extensive group size)
- A number of misunderstandings by the reviewers (e.g. loss divergence) were clarified

**Reviewer Scores:**

The reviewer KLKc may have raised their score due to significant clarifications while the others maintained.

---

### Decision · Program_Chairs · 2026-01-26

Reject